# Natural Motion for Energy Saving in Robotic and Mechatronic Systems

**Lorenzo Scalera** [1], **Ilaria Palomba** [1], **Erich Wehrle** [1,*], **Alessandro Gasparetto** [2]
**and Renato Vidoni** [1]

[1] Faculty of Science and Technology, Free University of Bozen-Bolzano, 39100 Bolzano, Italy
[2] Polytechnic Department of Engineering and Architecture, University of Udine, 33100 Udine, Italy
[*] Correspondence: erich.wehrle@unibz.it

**Abstract:** Energy saving in robotic and mechatronic systems is becoming an evermore important topic in both industry and academia. One strategy to reduce the energy consumption, especially for cyclic tasks, is exploiting natural motion. We define natural motion as the system response caused by the conversion of potential elastic energy into kinetic energy. This motion can be both a forced response assisted by a motor or a free response. The application of the natural motion concepts allows for energy saving in tasks characterized by repetitive or cyclic motion. This review paper proposes a classification of several approaches to natural motion, starting from the compliant elements and the actuators needed for its implementation. Then several approaches to natural motion are discussed based on the trajectory followed by the system, providing useful information to the researchers dealing with natural motion.

**Keywords:** natural motion; natural dynamics; energy saving; robotic system; trajectory planning; optimization

## 1. Introduction

Industrial robotic and mechatronic systems are required to have high energy efficiency, especially when high-speed continuous operations and high-volume production are needed [1]. Operating a robot or a mechatronic system at high speed produces significant losses at high velocities, as well as energy surpluses in the deceleration phases. These losses have repercussions on the amount of electric energy that is needed to operate the manufacturing system. Furthermore, increasing energy prices and environmental awareness encourage to reduce the power consumption. This is highlighted by the policy applied by the European Union, which aims to reduce the whole energy consumption up to 30% by 2030 [2]. Moreover, in the last years, the demand for industrial robots has accelerated due to the ongoing trend toward automation [3]. Therefore, their energy efficiency is becoming crucial since manufacturers are incentivized to install eco-friendly solutions for plants and production systems. For these reasons, engineers and researchers have been motivated to investigate and develop novel strategies to increase energy efficiency in industrial robots and mechatronic systems.

Several energy-saving methods for robotic and automatic systems can be found in the literature. In Reference [4], G. Carabin et al. present a classification of these methods, drawing a distinction between *hardware, software* and *mixed* approaches. In particular, hardware solutions include the implementation of new kinds of actuating systems, regenerative drives [5] and the design of lightweight manipulators [6–8]. The software approach is focused on time minimization, operations scheduling and trajectory optimization [9–13]. Mixed approaches rely upon the concurrent improvement of both hardware and software components of the automatic system. Among those, a particular method for enhancing energy efficiency is based on the concept of *natural motion*.

The definition of natural motion used here is as follows: a system response caused by the conversion of potential elastic energy into kinetic energy. Natural motion can be both a forced response assisted by a motor or a free response.

In the early 1980s, D. Koditschek defined the natural motion as the hope that the concurrent use of dynamic models and suitable control strategies might match the internal dynamics of the robot, affording more accurate performance with less effort [14,15]. The natural motion of a system is the one that occurs thanks to the transformation of its potential elastic energy into kinetic energy. The exploitation of this kind of motion allows to save energy in performing repetitive or cyclic motions, such as pick-and-place tasks [16,17] and legged locomotion (in walking and running robots) [18,19]. The more energy effective natural motion is the free response of the system: the unforced motion of a perturbed system that moves back to its equilibrium position. Indeed, in the ideal case of a conservative system, it keeps oscillate requiring only the initial energy to perturb its equilibrium. However, due to damping, friction forces and disturbances, the free response does not always coincide with a continuous oscillation that can be used in practice. Nevertheless, it is still possible to reduce energy consumption by exploiting the system natural motion by exciting it at resonance.

Systems working excited at resonance are quite common in several industrial applications. Examples are resonant conveyors, employed for material moving [20,21] and ultrasonic horns, used for cleaning, welding, cutting, machining and drilling [22,23]. To obtain a perfect match between the desired motion and the resonator natural dynamics, the resonant system is typically modified through changes in mass and/or stiffness parameters [24,25]. Modifications of system parameters are generally needed also to make a robot resonant [26], as well as to exploit its free response. Indeed, the vibratory behavior of traditional robotic systems can be neglected, since they are composed of links that are assumed to be rigid, which are connected through stiff joints and stiff actuators. Modifications are generally needed also dealing with lightweight robots that have elastic links, since link vibration could be too small compared with the displacements required by the task [27]. Large oscillations are typically obtained by installing lumped springs at the actuated joints. By properly tuning the compliance of the added springs, the natural dynamics of the robot can match the desired motion.

The aim of this review paper is to provide a classification and a discussion of existing technologies and methods to exploit natural motion for energy efficiency in robotic and mechatronic systems.

The remaining of this paper is organized as follows: Section 2 describes how a robotic system can be modified to exploit the natural motion by adding elasticity. Section 3 provides a classification and a description of the different methods developed on the basis of the trajectory followed by the robotic system. In Section 4 the optimal design problems in natural motion are analyzed, since the implementation of natural motion is often related to the definition of an optimization problem. Then, Section 5 analyzes some open issues and critical points in the field of natural motion. Finally, Section 6 summarizes the conclusions of this survey together with possible future developments.

## 2. Mechanical Design for Natural Motion

The natural motion can only be applied to vibratory dynamic systems. This means that the system has to be capable of conveying kinetic energy and of storing potential energy. Every robotic system is capable of conveying kinetic energy thanks to its mass and inertia but it is not always equipped with compliance elements to store elastic energy, that hence have to be added. The concept of adding springs or additional elements to a robotic system in order to reduce the efforts of the actuators is first adopted in gravity-balancing techniques [28,29]. Examples of gravity balancing techniques can be found in References [30–35]. These methods are proposed to compensate the input effort required to move the links of a pick-and-place robot and to reduce energy consumption in slow movements. However, static-balancing techniques cannot be implemented in high-speed manipulators in which the dynamic effects are not negligible. To exploit the natural motion, the elastic elements are installed at robot joints. The possible connections between elastic elements and joints are three: *serial, parallel*

and their combination *serial and parallel*. These three configurations are analyzed in the following. Furthermore, a comparison of these configurations is discussed at the end of this section.

### 2.1. Serial Compliance Elements

Serial elastic elements can be added to a robotic system by connecting an electric motor to a joint by means of an elastic element (see Figure 1a). Such an arrangement is not simply obtained by adding a spring between them but instead by substituting stiff actuators with compliant ones. Series elastic actuators (SEA), first introduced by G.A. Pratt in Reference [36], are constituted of a linear compliant element between a high impedance actuator and the load. Their use is demonstrated to be beneficial in compliant actuation, energy storage, interaction safety, in the absorption of contact shocks and in the reduction of the peak forces due to the impacts in bipedal walking robots [36,37]. Examples of SEA for minimizing energy consumption can be found in References [38,39]. In these works, a convex optimization problem is presented for the design of a SEA capable of energy regeneration such that energy consumption and peak power are minimized for arbitrary periodic reference trajectories.

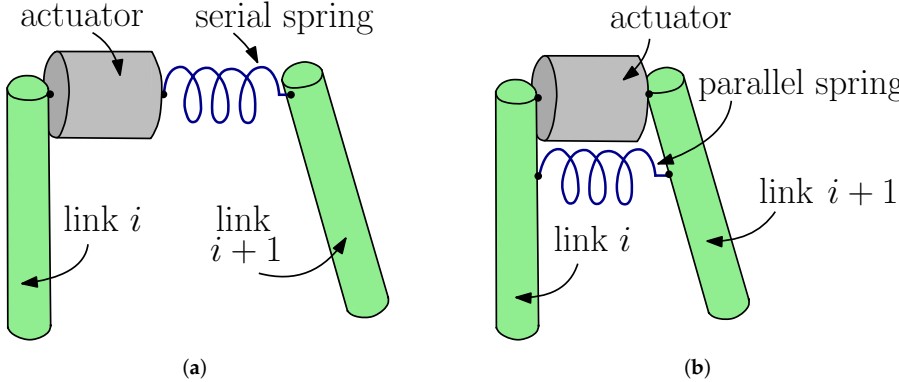

**Figure 1.** Compliance configurations: (**a**) spring in series and (**b**) spring in parallel.

The main drawback of SEA is that the stiffness is fixed and cannot be changed during motion, thus limiting the level of compliance to adapt for several tasks. To overcome this problem, a novel class of actuators was proposed: the variable stiffness actuators (VSA). These actuators consist of a motor connected to the output link by a spring in series, whose stiffness is variable. Therefore, the stiffness can be properly controlled to reduce energy requirements during the execution of repetitive tasks, as explained in Reference [40]. Reviews on variable stiffness actuators can be found in References [41–44], whereas a comparison of several design of VSA based on spring pretension is illustrated in Reference [45].

VSA were first introduced to decrease contact shocks, to enhance soft collisions in human-robot interaction [46,47] and to efficiently actuate legged locomotion systems [48,49]. Furthermore, they were employed to decrease energy consumption in cyclic operations of robotic systems. An example is given by the actuator with adjustable stiffness proposed by A. Jafari et al. [50–52].

### 2.2. Parallel Compliance Elements

Compliant elements can be connected in parallel to the main actuators as shown in Figure 1b. Although some parallel elastic actuators can be found in the literature [53–55], it is not necessary to replace the original actuators to install parallel springs. For example, M. Iwamura et al. equipped a serial robot with two linear springs placed at joints, between neighboring links and a special spring holder [56]. In this way, they overcome the difficulties in adjusting the stiffness value and the mounting positions of torsional springs.

The research in this field mainly addresses the development of mechanisms to realize variable stiffness springs [57–59] and non-linear off-the-shelf springs [60–63]. Indeed, non-linear stiffness

allows a larger energy saving in robots, since actuators torques and end-effector trajectory are always related by non-linear behavior, as demonstrated in References [58,64,65].

A mechanism to change spring stiffness is proposed by M. Uemura et al. in Reference [57]. It consists of a sliding screw with a self-lock function and a linear spring. The self-lock function guarantees that a constant elastic value is kept when the motor of the variable elastic mechanism does not exert a torque. The linear spring is attached by one end to the actuated link of the system and by the other to a point on the lead screw mechanism. By changing the position of the lead screw the length of moment arm of the elastic force exerted by the linear spring changes and hence the equivalent torsional stiffness.

R. Nasiri et al. realize a parallel variable torsional spring by means of two linear springs and two worm-gear motors [58]. One end of the two linear springs is connected to a point of the actuated link, whereas the other end to the worm-gear motor. By independently controlling the strain-length of the springs with the two motors, an arbitrary compliance profile at the actuated joint can be obtained.

A mechanism capable of realizing a constant non-linear torsional springs is presented by N. Schmit and M. Okada [60,61]. The mechanism consists of a linear spring connected to a cable wound around a non-circular spool. A non-circular spool (or variable radius drum) is characterized by the variation of the spool radius along its profile [66]. In References [60,61], the spring mechanism is attached to each actuated joint, with respect to which the linear spring behaves as a non-linear rotational spring with a described torque profile given by the shape of the spool (Figure 2). A similar design is proposed also by B. Kim and A.D. Deshpande [67], who additionally design an antagonistic spring configuration for bilateral torque generation.

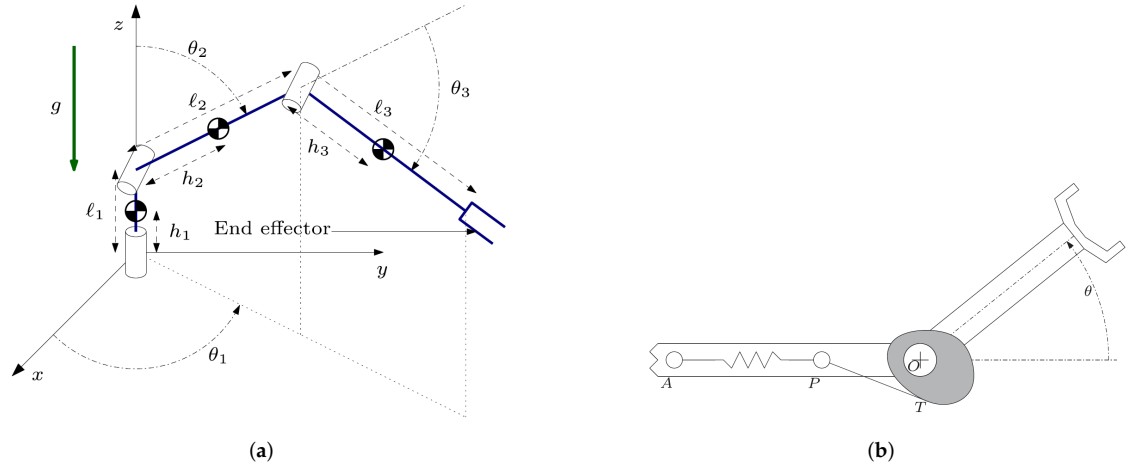

(**a**)            (**b**)

**Figure 2.** (**a**) The 3-DOF serial manipulator employed in the simulations by N. Schmit and M. Okada and (**b**) the design of the proposed non-linear spring mechanism [61].

H.J. Bidgoly et al. [63] realize a non-linear torsional spring with an arbitrary stiffness profile by combining a linear spring and a non-linear transmission mechanism. The last consists of a non-circular cam connected to the actuated joint and a roller, which moves along the outer circumference of the cam. The stretched linear spring is hinged to the centers of the cam and the roller. The desired torque-angle profile is obtained by properly designing the shape of the cam.

Another system to attach springs in parallel with the actuator is proposed by M. Plooij and M. Wisse [62]. A parallel spring mechanism converts the linear stiffness of a linear spring in an equivalent torsional non-linear stiffness. The mechanism consists of two pulleys of different size connected through a timing belt. The larger pulley is attached to the actuated link. The two ends of the spring are connected to two points placed on the outer circumference of the smaller pulley and of the larger one, respectively. In this way, the spring is non-linearly stretched with respect to the rotation of the link. One peculiarity of this mechanism is that it has two different configurations in which the

elastic energy of the spring is null. This means that if the mechanism is properly designed for a specific pick-and-place operation, the actuators does not have to counteract the spring during the task.

It is worth noting that there are two mounting possibilities for parallel springs, that is, they can be connected to just one joint, or they can span over two joints. In the first arrangement, largely adopted, springs are called mono-articular, whereas in the second, bi-articular (Figure 3). Bi-articular springs take inspiration from biological studies [68]—bi-articular muscles actuate human limbs as well as paws of birds, reptile and insects. Non-linear bi-articular springs are designed for example by B. Kim and A.D. Deshpande [67], by means of pulley-cable mechanisms and antagonist linear springs. Bi-articular springs are mostly employed in the realization of walking robot [69–71]. The effectiveness of such springs in serial manipulators to perform pick-and-place operations are investigated by G. Lu et al. [72] and H.J. Bidgoly et al. [73], with different conclusions. In the case of a variable linear bi-articular spring added on a SCARA robot, G. Lu et al. [72] conclude that such a spring alone cannot effectively save energy. Conversely, H.J. Bidgoly et al. [73] assert that bi-articular spring contribute more in the actuation cost minimization with respect to mono-articular springs, in the case of a redundant 4-DOF serial manipulator with non-linear mono- and bi-articular springs.

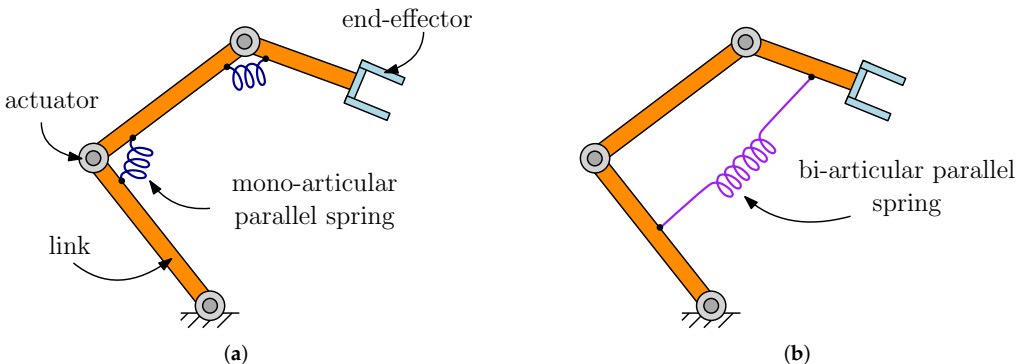

**Figure 3.** (**a**) Mono-articular and (**b**) bi-articular parallel springs mounted on a robotic manipulator.

### 2.3. Serial and Parallel Compliance Elements

In the context of natural motion, some actuators are developed combining both serial and parallel compliant elements. In these systems, the serial compliance is employed for impact mitigation, whereas the parallel one is adopted for efficient energy storage.

In Reference [74], a mixed series-parallel approach is presented by G. Mathijssen et al. Multiple series elastic actuation branches are placed in parallel, each engaged depending on torque requirements. This design leads to a significant torque effort reduction, as well as an increased output torque range, compared to traditional stiff or SEA configurations.

This actuation design is adopted by N.G. Tsagarakis et al. [75] and by W. Roozing et al. [76]. A novel asymmetric actuation scheme that consists of two actuation branches that transfer their power to a single joint through two compliant elements with diverse stiffness and storage capacity properties is presented. The serial branch is used for impact absorption and it is connected to the main actuator, whereas the parallel branch is adopted for its large potential energy storage capability in cyclic tasks.

### 2.4. Discussion of Elastic Elements Configurations

Both serial and parallel configurations can be implemented with springs of *variable* or *non-variable* stiffness, as well as *linear* or *non-linear* behavior. Variable compliance allows the *online* tuning of spring parameters. Therefore, control systems that act on the stiffness to perform varying operations and to change system dynamics can be adopted, as in References [16,58,77,78]. As a drawback, variable compliance springs need a more complex actuator design and sensory feedback. On the contrary, configurations based on non-variable stiffness are easier to implement, the springs can be designed and tuned *offline* but are suitable only for non-varying tasks and operations.

The serial configuration (adopted for example in References [42,50,79]) provides the actuator with a compliance that can be employed to decrease contact shocks and to reduce force peaks due to impacts, for example, In human-robot interaction [46]. On the other hand, the parallel configuration [65,80–82] results in simpler mechanism and mathematics formulation [78]. Parallel compliance does not enlarge the configuration space of the robotic system and has a well-posed quadratic cost function for energy consumption minimization with respect to the compliance coefficients [83]. Moreover, the serial arrangement of springs and motors limits the operational speed due to uncontrolled robot deflection when performing high-speed tasks.

Comparisons between serial and parallel configurations can be found in References [84–88]. T. Verstraten et al. in Reference [87], provide a comparison between stiff actuators, parallel elastic actuators and serial elastic actuators in terms of power as well as mechanical and electrical energy consumption. As test case, a sinusoidal motion is imposed to a pendulum load. By means of simulations they demonstrate that, if the stiffness of the elastic actuators is properly tuned, it is possible to reduce energy up to 78% compared to rigid actuator using serial elastic actuators and up to 20% using the parallel ones.

Although the results of this study demonstrate that a serial spring arrangement outperforms the parallel one from an energetic point of view, the majority of the works dealing with natural motion adopt parallel springs because of their easiness of installation and control. An evidence of this is provided by Table 1, where the spring installation configuration adopted in the reviewed papers are highlighted.

**Table 1.** Literature on natural motion in robotic and mechatronic systems.

| Ref. | 1st Author | Year | Path and Trajectory Planning | Compliance of Elastic Elements |
|---|---|---|---|---|
| **Defined trajectory** | | | | |
| [89] | Boscariol | 2017 | cycloidal, 5-degree polynomial trajectory | linear parallel |
| [90] | Carabin | 2019 | double-S speed profile | linear parallel |
| [40] | Fabian | 2018 | harmonic trajectory | variable parallel |
| [16] | Goya | 2012 | harmonic trajectory | variable parallel (mono- and bi-articular) |
| [91] | Hill | 2019 | adjustment of limit cycle | variable parallel |
| [65] | Khoramshahi | 2014 | circular path in the operational space | non-linear variable parallel |
| [92] | Lu | 2012 | desired motion given | linear parallel |
| [72] | Lu | 2012 | harmonic trajectory | variable parallel |
| [93] | Matsusaka | 2016 | desired motion given | variable parallel |
| [94] | Matsusaka | 2016 | harmonic trajectory | variable parallel |
| [58] | Nasiri | 2017 | harmonic trajectory | linear and non-linear variable parallel |
| [64] | Scheilen | 2005 | harmonic trajectory | linear and non-linear parallel |
| [95] | Scheilen | 2005 | harmonic trajectory | linear parallel |
| [96] | Shushtari | 2017 | sinusoidal path given | non-linear parallel |
| [77] | Uemura | 2007 | harmonic trajectory | variable parallel |
| [97] | Uemura | 2008 | harmonic trajectory | variable parallel |
| [98] | Uemura | 2008 | harmonic trajectory | variable parallel |
| [99] | Uemura | 2009 | harmonic trajectory | variable parallel |
| [78] | Uemura | 2014 | harmonic trajectory | variable parallel |
| [86] | Velasco | 2013 | harmonic trajectory | linear serial/parallel |
| [79] | Velasco | 2015 | squared trajectory | linear and variable serial |
| **Optimized trajectory** | | | | |
| [73] | Bidgoly | 2017 | cyclic end-effector path given | non-linear parallel (mono- and bi-articular) |
| [80] | Hill | 2018 | polynomial trajectories for robot and variable stiffness springs | variable parallel |
| [56] | Iwamura | 2011 | trajectory optimization | linear parallel |
| [100] | Iwamura | 2016 | trajectory optimization | linear parallel |
| [82] | Mirz | 2018 | optimal control theory path planning | linear parallel |
| [101] | Scheilen | 2009 | trajectory optimization | linear parallel |
| [102] | Schmit | 2012 | trajectory discretized in the joint space | non-linear parallel |
| [61] | Schmit | 2013 | trajectory discretized in the joint space | non-linear parallel |

**Table 1.** *Cont.*

| Ref. | 1st Author | Year | Path and Trajectory Planning | Compliance of Elastic Elements |
|------|-----------|------|------------------------------|--------------------------------|
| **Free-vibration response** | | | | |
| [103] | Barreto | 2017 | defined by the multibody simulator | linear parallel |
| [81] | Barreto | 2018 | defined by the multibody simulator | linear parallel |
| [104] | Kashiri | 2017 | periodic motion defined by the natural frequency | variable serial |
| **Periodic trajectory learning** | | | | |
| [105] | Khoramshahi | 2017 | harmonic trajectory | linear/variable parallel |
| [83] | Nasiri | 2016 | sinusoidal, polynomial and sign basis functions | linear parallel |
| [106] | Noorani | 2013 | motion defined by the pendulum | linear, applied at pendulum tip |

## 3. Design of Desired Natural Dynamics

The concept of natural motion can be exploited to perform tasks that are compatible with the natural dynamics of a mechanical systems, that is, cycle tasks. For this reason, such a concept is mainly adopted for locomotion of legged robots or, in the industrial field, for pick-and-place and palletizing tasks.

A pick-and-place (or palletizing) task has strict requirements on the positions where objects are located and the corresponding velocities. Specifications on the task time and the path could potentially be relaxed constraints.

Conversely, walking robots do not need strict requirements for the gait. In this case, if the robotic system is already equipped with elastic elements, the desired task (the robot gait) can be adapted to the system characteristics, that is, The task can be performed by adopting the trajectory naturally generated by the system. Such an approach can be referred to as *natural dynamic exploitation* [96]. Generally speaking, when there are requirements on the task to execute, it is always necessary to modify the system to fulfill them. The adaptation of the system to perform a given periodic task can be indicated as *natural dynamic modification* [73].

As discussed in Section 1, the modifications of a robotic system to exploit the natural motion rely on spring additions. Therefore, the problem can be reformulated in this way: how should the spring parameters be determined in order to fulfill the task requirements exploiting system dynamics? Such a question has not a unique and straightforward answer. Requirements on task are typically given in terms of positions, velocities and time. Dynamic models of robotic systems depend non-linearly on the first two. Additionally, to find an analytical solution for the motion equations is very difficult, if not impossible, and hence also the setting on the time requirement is impossible, unless simplifications (such as linearization) are adopted or assumptions on the system response law are made. Therefore, since the optimal spring depends on the trajectory and *vice versa*, we classify the methods concerning with natural motion according to the trajectory followed by the system:

- *Defined trajectory*. A feasible trajectory for the desired task (typically harmonic) is imposed and the spring parameters are optimized to minimize a given objective function related to energy consumption;
- *Optimized trajectory*. The spring parameters and the system trajectories are concurrently optimized thanks to a parametric representation of both or by adopting the optimal control theory;
- *Free-vibration response*. The trajectory is not imposed. The optimal spring parameters are identified so that the free response of the system fulfills task requirements. Such a result can be obtained by means of linearized dynamic models or multibody simulators;
- *Periodic trajectory learning*. The robotic system is not modified. The forced response at resonance of the system is learned by means of proper tools and used as reference trajectory.

These four categories are analyzed and discussed in the following.

### *3.1. Defined Trajectory*

Most of the works dealing with natural motion adopt a fixed pre-defined trajectory, which usually consists of an harmonic motion law. In the case of defined trajectory, the correct springs parameters to be added in the system can be determined with different strategies: control-based methods, graphical approaches or optimization strategies. These three approaches are discussed in the following.

### 3.1.1. Control-Based Methods

M. Uemura et al. in Reference [77], employ a control method based on linear stiffness adjustment to reduce the actuator torque needed to follow a desired harmonic trajectory with a serial link system. The proposed adjustment law for the parallel spring stiffness depends on the angle and angular velocity tracking errors. Such a law is added to the feedback control system: when the tracking errors become smaller by the stiffness adjustment, the feedback terms become smaller as well. In this manner, the actuator torques are reduced. The variability of the linear stiffness is exploited just to tune the system at different desired frequencies of the harmonic trajectory. In particular, the stiffness varies just at the beginning of the motion, during the tuning phase, before converging to a constant value.

H. Goya et al. in Reference [16], experimentally validated the stiffness adaptation control on a 3-DOF SCARA robot equipped with two adjustable parallel springs on the two revolute joints. The desired start and end points are determined in the task-oriented coordinates and the corresponding points in joint angle coordinates through inverse kinematics. The start and end points in the joint space are connected through an harmonic trajectory with a given period and amplitude equal to the distance between the start (end) point and the equilibrium point. The equilibrium position of the elastic element is set at the middle point between the start point and the end one in the joint coordinates. The springs are tuned to perform the given motion between two points, as discussed in Reference [77].

An alternative method to perform multi-point trajectories and concurrently exploit the adaptive stiffness control to minimize the actuator efforts is proposed by K. Matsusaka et al. in Reference [93]. In this work, it is assumed that the elasticity can be instantaneously changed when the spring is in equilibrium. In this way, the amplitude of the oscillation can be modified without affecting the potential energy stored by the elastic element. With respect to the strategy proposed by H. Goya et al. based on the feedback control method [16], this approach allows to increase the number of pick-and-place points and to reduced the energy consumption by 39%, as proved experimentally. Furthermore, M. Uemura et al. prove in Reference [98] the effectiveness of the stiffness adaptation controller [77] on a 1-DOF system. Multi-frequency harmonic trajectories are tracked, while minimizing the norm of the required torque.

Variable elastic elements and the control proposed in Reference [77] are used by K. Matsusaka et al. in Reference [94] to improve the energy-efficiency of a 2-DOF robot in palletizing task. For such a task an obstacle-avoidance trajectory is proposed consisting in moving both the joint with harmonic trajectories having an angular frequency ratio 2:1. Additionally, to increase the energy savings, the authors suggest to provide the system not only with variable stiffness springs but also with linear constant springs. In this way, the gravitational force can be counteracted.

An improvement of the adaptation stiffness controller in terms of energy saving is obtained in Reference [97], by combining such a controller with a delay feedback control. However, although the new approach allows to move the system almost without any actuation force, the delay feedback control modifies the desired trajectory. Therefore, the resulting motion can be significantly different from the desired one.

A solution to improve the energy savings of the adaptive stiffness control and achieving a good trajectory tracking control is proposed in Reference [99] and experimentally validated in Reference [78]. The new control method combines the stiffness adaptation and the iterative learning control and can be applied to multi-joint robots. The proposed control optimizes the stiffness of elastic elements installed in parallel at each joint in order to save energy and to track a desired multi-frequency harmonic trajectory.

An online adaptation method suitable for non-linear compliance acting in parallel with actuators is presented in Reference [58]. The method aims at minimizing actuation forces of multi-joint robots performing given cyclic tasks. Such a method adds an adaptation rule in parallel to the closed-loop control in order to minimize the squared actuation forces. The parameters that allow to minimize the force are the adaptable coefficients used in the compliance definition. The compliance of each actuated joint is defined as a multi-basis non-linear compliance, that is, as the sum of the products between a coefficient and a smooth basis function defined over the joint position. In particular, the compliance structure is defined by the basis functions that are decided a priori and fixed, whereas the coefficients are adaptable and used to minimize the cost function. By choosing a proper set of basis functions (e.g., polynomials), the compliance force acts as a general function approximator. Hence the elastic force has more flexibility to compensate the actuation torques, which are typically in a non-linear relationship with the joint angles. Similar to the other methods discussed up to now and based on control system, this method does not require any knowledge of the controlled system or of the dynamical equations of the robot.

A recursive algorithm to adjust the configuration of a variable spring actuator for a given trajectory is proposed in Reference [40]. Unlike the previous methods to tune variable stiffness, such an algorithm does not optimize the mechanical output. It directly reduces the input electrical energy requirements of the system during the execution of repetitive tasks without requiring a precise knowledge of the controlled system. It is based on the gradient descent optimization algorithm [107]. Basically, it expresses the objective function (the total electrical input energy of the actuator) as a convex function in the design parameters (i.e., The spring stiffness). An iterative process finds the values of the configuration parameters that minimize the objective function in a repetitive task performed by the actuator. The inputs of the algorithm are real-time measurements or estimations of the objective function, the physical limits of the design parameters and the periodicity of the task.

All the control-based approaches discussed up to now do not require the knowledge of the system dynamic model. Such an aspect not only simplifies their implementation but makes these approaches more robust to typically unmodeled physical phenomena as, for example, friction or noise, as experimentally demonstrated in References [40,99]. In Reference [99], the validity of the method is proved also in the presence of static friction, Coulomb friction and backlash. In Reference [40], the robustness of the algorithm to the variation of the objective function due to the changes in the operation condition, perturbations and signal noise is demonstrated. On the other hand, all these methods, although exploit variable stiffness springs, are intended for repetitive tasks with a relatively low rate of change of the stiffness. Therefore, the convergence to the optimal stiffness value is not instantaneous but can last several cycles.

A method proposed by W. Schiehlen and N. Guse [64] takes advantage of the inverse dynamic model and considers constant linear springs mounted in parallel with the actuators. A control based on limit cycle to reduce energy consumption in robotic system performing periodic tasks is proposed. Given the period of the task and the desired trajectory of the system, this is then adapted to match the limit cycle of the mechanical system at best.

The definition of the desired trajectory includes the identification of the boundary conditions for the state of the motion as well (i.e., position and velocity of the start and end points). By adopting a modified shooting method [108], the values of the stiffness and the neutral position of the parallel spring are adjusted to meet the boundary conditions of the system featuring a limit cycle close to the desired trajectory. A low energy control is sufficient to force the system to follow the desired trajectory. With reference to a 2-DOF assembly robot performing an harmonic motion law, the limit cycle of the system can be correctly adjusted by properly choosing the coefficient of linear springs. In the case of an arbitrary, non-harmonic motion law, such as a piece-wise constant acceleration trajectory, non-linear springs are needed to correctly adjust the limit cycle to the desired trajectory. In Reference [91], R.B. Hill et al. extend the method proposed by W. Schiehlen and N. Guse to multiple-point trajectory. In this case, variable stiffness springs are employed. The spring parameters can be tuned to force the

limit cycle to converge to the desired pseudo-periodic trajectory between every two consecutive points of the multiple-point motion.

G. Lu et al. propose a control method starting from a linearized model of a 2-DOF planar manipulator, lying in the horizontal plane [72]. The purpose is to save energy in performing a given harmonic trajectory. The controller adaptively tunes the stiffness of parallel springs, while compensating for viscosity effect. The adaptive control law for the stiffness is proportional to the errors between the actual and the desired joint velocities and the difference between the actual joint position and its equilibrium one. The same authors propose an inertia adaptive control for energy savings in Reference [92]. This work starts from a 1-DOF system already equipped with a constant linear spring and add a movable mass to tune and adapt the eigenfrequency of the system. In this way, the frequency of the desired harmonic trajectory can be reached.

Another control-based approach to employ stiffness variability for energy efficiency during a task is proposed by A. Velasco et al. in Reference [79]. The method aims at determining the optimal stiffness profile of serial springs that minimizes the energy consumption of the mechanical system, performing a given task. Since changing the actuator stiffness has an energetic cost, the authors add such a cost to the objective function (squared mechanical torque). By exploiting the cost function and the motion equations of the actuated system, an analytical solution for optimal stiffness as function of the desired trajectory is found. In particular, the total time is subdivided in intervals and an optimal stiffness profile, that can be constant or variable, is determined for each sub-interval. Simulations with squared trajectories with different amplitudes and frequencies show that the optimal spring profile (i.e., constant or variable) depends on the reference trajectory. Constant stiffness is preferred for low amplitudes and low frequencies trajectories, whereas variable stiffness is suggested for high frequency motion laws.

### 3.1.2. Methods Based on Force-Displacement Graphs

W. Schiehlen and N. Guse propose in Reference [95] an alternative method to that in Reference [64] for determining the spring design parameters that minimize the actuators work. The required control forces or torques are computed by means of the inverse dynamics and the desired trajectory. Such forces are then expressed as function of the positions of all the bodies and fitted by means of polynomials. If a linear function is used as a curve fit, which represents a linear spring, the spring coefficient is the slope of this curve fit and the axis intercept is the spring fastening. The use of curve fitting is equivalent to a minimization task with inequality constraints, since the stiffness coefficient is always positive.

The exploitation of the force-displacement graph is suggested also by M. Khoramshi et al. [65]. In particular, they compute the absolute work along the force-displacement graph and derive the value of the parallel spring stiffness that minimize it. Such a strategy results in a non-linear stiffness profile capable of improving the energy efficiency of the system. Indeed, with the adding of parallel non-linear compliance, the resulting force-displacement graph is lined up around the horizontal axis (i.e., null force).

### 3.1.3. Methods Based on Offline Optimization

An analytical method for the offline computation of the optimal parallel compliant elements and the frequency of the reference trajectories for serial manipulators performing cyclic tasks is presented by M. Shushtari et al. [96]. A representation of the compliant elements similar to those used by R. Nasiri et al. [58] is adopted. In particular, they take into account a multi-basis representation consisting in the product between a compliance coefficient and a basis function dependent on the joint coordinates. Such a representation leads to a cost function (squared mechanical torques multiplied by a weighting matrix) that is a quadratic function with respect to the compliance coefficients and quartic with respect to the task frequency. This function can be analytically solved. In order to address the multi-task case, M. Shushtari et al. propose a weighted sum of the original cost functions defined over the tasks.

P. Boscariol et al. [89] set up a constrained optimization problem to find the optimal stiffness value and placement of a linear spring to be added in parallel with the joint actuator of a 1-DOF system for reducing the peak torque requirement (Figure 4). The authors also shown the effects of the joint trajectory (cycloidal motion trajectory and 5th order polynomial with null initial and final acceleration) on the optimization results.

In Reference [90], G. Carabin et al. propose a methodology to reduce the electrical energy consumption in a Delta-2 robot by concurrently exploiting energy recuperation drive axles and torsional springs mounted in parallel to the actuators. Figure 5 reports the kinematic diagram and the electrical schematic of the Delta-2 robot. An optimization-based design method determines the stiffness of the two torsional springs and their equilibrium positions. The energy consumption performing cyclic pick-and-place operations, with a predefined trajectory (double-S speed profile in the workspace) is minimized.

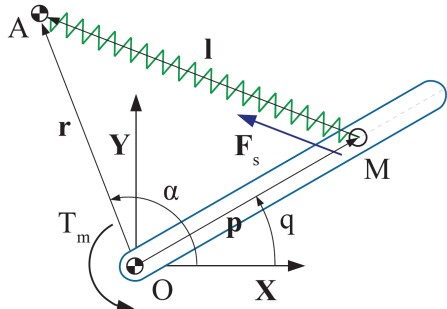

**Figure 4.** The 1-DOF mechanism with linear spring used by P. Boscariol et al. [89].

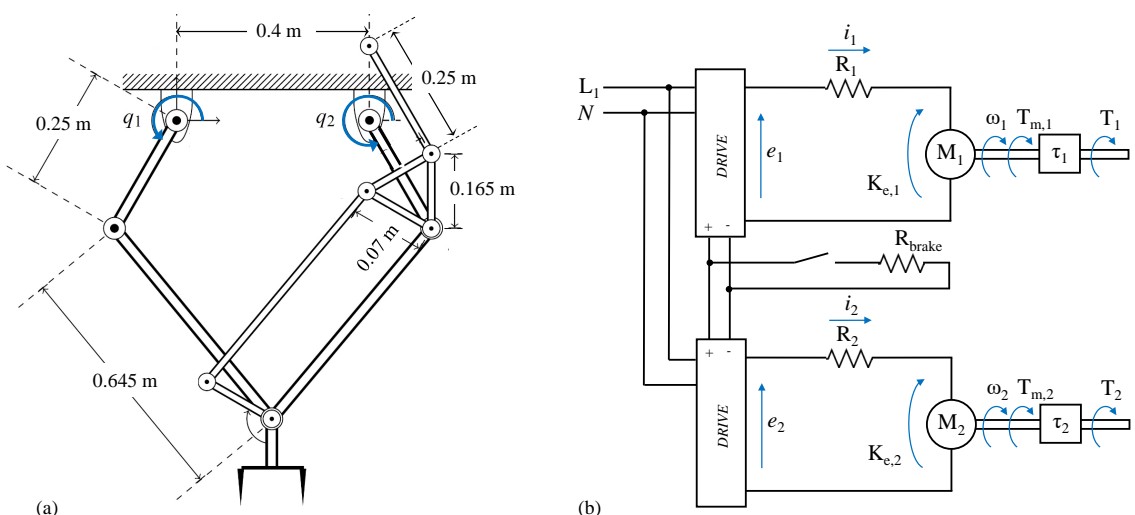

**Figure 5.** (**a**) The Delta-2 robot and (**b**) the electrical schematic of its energy recuperation drive axles employed by G. Carabin et al. [90].

A. Velasco et al. in Reference [86], determine the optimal stiffness value and spring preload such that a given cost functional is minimized. In particular, they consider the influence on the optimal values of different aspects: spring placements (i.e., serial or parallel actuator); parameters of a given harmonic trajectory (amplitude and frequency); cost function (squared mechanical torque or squared mechanical power). From such a study, it results that, if parallel spring are employed, the optimal values for spring stiffness and preload can be analytically found regardless the cost function employed. Conversely, for serial springs an analytical solution exists only if the cost function is the squared torque. As far as the effects of the trajectory parameters on the energy savings are concerned, the authors show that, if serial springs are employed, the savings depend also on the cost function adopted. On the

other hand, parallel springs have the same trends independently from the cost function. In particular, it results in the parallel springs being more convenient for small amplitudes at low frequency or large amplitudes at high frequency. The use of serial springs is more convenient for small amplitudes at high frequency or for large amplitudes at low frequency, if mechanical torque is considered or for high frequencies independently from the amplitude, if mechanical power is considered.

### 3.2. Optimized Trajectory

Another strategy to improve the energetic efficiency of the system consists of the concurrent optimization of trajectory and spring stiffness. Two are the most common methodologies to reach such a goal—methods based on the optimal control theory, which find an optimal control law that minimizes a given cost function, and methods that parametrize the trajectory by means of basis functions.

### 3.2.1. Methods Based on the Optimal Control Theory

W. Schiehlen and M. Iwamura [101] simultaneously optimize the constant linear spring stiffness (mounted in parallel with the actuator) and the joint trajectories with respect to the energy consumption, taking advantage of the optimal control theory. Following their approach, the time to execute the task is not given but it is optimized as well. Indeed, they define a relationship between the consumed energy and the operating time and the optimal trajectory, finding a condition for the operating time to be optimal. Then, the optimal design for the springs is derived accordingly to such a time. The aforementioned relationship as well as all the optimal solutions are derived starting from linearized equations of motion. An analytical solution is provided for the optimal trajectory (which results in harmonic motions in modal coordinates), the operating time and the spring equilibrium position (corresponding to the middle point between the initial and final desired points). Conversely, the optimal stiffness is numerically found by means of an iterative method. Although the authors verify the correctness of the analytical solution proposed by comparing it with the numerical one based on the non-linear dynamics, their method is valid as long as the linearization assumptions are respected, that is, The centrifugal and Coriolis forces are negligible. This means that such a method is suitable if fairly strong springs or long operating time are adopted. A manipulator prototype, respecting the validity assumptions, is used by M. Iwamura et al. [100] for the experimental validation of this approach. The method proposed by W. Schiehlen and M. Iwamura [101] is extended by the same authors to systems working under gravity in [56].

The optimal control theory is exploited by C. Mirz et al. [82] to reduce energy consumption in parallel kinematic manipulators by means of linear torsional springs mounted in parallel to the motors. The power consumption of the drives is selected as cost function to calculate the optimal trajectory to travel between given initial and final positions in a fixed cycle time with minimum energy. By applying the Pontryagin's minimum principle, the cost function is transformed in a two points boundary value problem solved numerically by means of finite difference method. Although the characteristics of the elastic elements are determined for one specific initial and final position and a given cycle time, simulations showed satisfactory results in terms of energy savings also performing pick-and-place operations between 200 different positions arranged in a square about the initial and final points.

### 3.2.2. Methods Based on Parametrization through Basis Functions

N. Schmit and M. Okada et al. [102] minimize the actuator torques by simultaneously designing the robot trajectory and the torque profiles of the non-linear parallel springs located at each joint. The desired time interval to perform the task is divided into a certain number of equal sub-intervals, based on which a third-degree Hermite interpolation is used to parameterize both the joint trajectory and the spring torque profiles. By expressing the spring torques as function of the trajectory, the position and velocity of each joint at the nodes become the only design parameters. The optimal joint trajectory is found numerically minimizing a cost function composed of three terms—a term evaluating the actuator torques, a term evaluating the improvement due to the contribution of the non-linear springs and a

term that weights the non-linearity of the profiles to guarantee their technical feasibility. Once the optimal joint trajectory is found, the optimum spring is designed as function of the optimal joint trajectory with a closed-form solution. However, the resulting springs may exhibit negative stiffness. Such an issue is overcome in Reference [61], where constraints to impose positive stiffness to given joint coordinates are added to the optimization problem.

The use of polynomials to parameterize the joint trajectory and spring torque profiles is suggested by H.J. Bidgoly et al. in Reference [73]. In contrast to Reference [102], the authors adopt one polynomial (with a degree to be determined by the designer) for all the time interval. Also in this case, the optimal spring are computed analytically, whereas the joint trajectories are numerically optimized: the spring torque profiles are polynomial functions of the joint trajectories, which, in turn, are polynomial functions of time. The optimal trajectories are found by solving a constrained optimization problem whose cost function takes into account actuator torques and realization complexity of the non-linear springs. The possibility to obtain an optimal trajectory, in the meaning that it is very close to the robot natural behavior, is increased by the assumption that the system is redundant with respect to the task, so that infinite solutions of inverse kinematics are possible. However, although the authors show that very good results can be obtained in terms of actuation minimization with their methodology, they do not provide an evidence that kinematic redundancy has advantages over the non-redundant case.

R.B. Hill et al. in References [80,91], propose a method to increase the energy efficiency of parallel robots, by concurrently optimizing the joint trajectory and the control law of variable springs mounted in parallel with the system actuators. Figure 6 shows the adopted five-bar mechanism performing a pick-and-place operation and the power transmission system of variable stiffness springs. The authors do not focus on the spring profiles but on the joint trajectories of the spring actuators so that a classical position controller can be used without a feedback measure of the stiffness. Similar to Reference [102], the total time to perform the task is subdivided in a certain number of equal subinterval, defined by the number of intermediate points (via points) chosen between the initial and the final positions for both the robot joints and spring motors. The via points are interconnected thanks to an expression defined by means of four polynomials, each one function of position, velocity and acceleration of the considered link and of the time, respectively. The explicit form of each polynomial is obtained by means of a heuristic approach: different functions are experimentally tested until the best polynomial form in terms of consumed energy during the motion is obtained. The optimal trajectory is defined looking for the position, velocity and acceleration of the intermediate points of both the robot joints and spring motors that minimize the energetic losses of the entire actuation chain (robot joint motors and variable stiffness motors). The larger advantage of the method proposed by R.B. Hill et al. in References [80,91] over the other discussed in this section, is the possibility of planning and optimizing multiple-point trajectories.

### 3.3. Free-Vibration Response

In the context of natural motion, few methods exploit the free-vibration response of the robotic system for performing a desired task. N. Kashiri et al. [104] propose a method to exploit the natural dynamics of serial robots driven by adjustable compliant actuators mounted in series. A feedback linearization control is used to linearize the system dynamics and to generate modally decoupled limit cycles. Having decoupled the system dynamics, it is possible to analytically write the response of the closed-loop system and to set its natural frequencies to the target values by properly tuning the compliant elements. The control scheme generates a smooth reference link position to excite an individual mode, as well as a combinations of modes allowing for the desired periodic motion with a minimal energy expenditure.

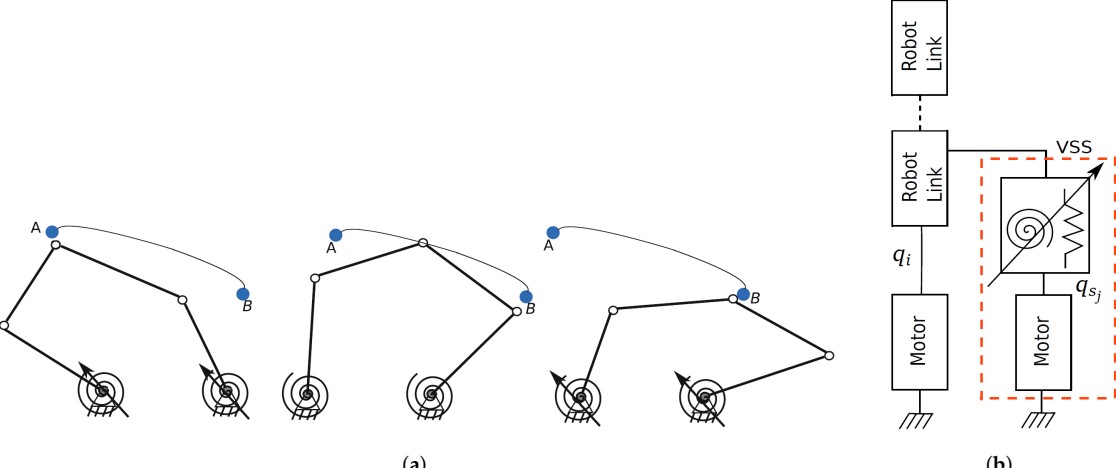

(**a**)                                                                                              (**b**)

**Figure 6.** (**a**) Pick-and-place operation performed by a five-bar mechanism equipped with parallel variable stiffness springs (VSS). (**b**) Scheme of the VSS power transmission system employed by R.B. Hill et al. [91], where $q_i$ represents the *i*-th robot joint coordinate, whereas $q_{sj}$ the *j*-th variable stiffness spring coordinate.

J.P. Barreto et al. exploit the free response of a five bar linkage mechanism [103] and of a Delta robot (Figure 7) [81] to move the system between two given points for a pick-and-place operation in a prescribed time. To reach such a result, a set of springs with optimized parameters (stiffness and equilibrium positions) are added in parallel to the motors. The optimal spring parameters are found by solving the system direct dynamics with the aid of a multibody simulator. By imposing the initial desired position of the system (pick position) and taking a first guess for the spring parameters, the simulator computes the system positions and velocities after the task time, which are compared to the desired ones (place positions and velocities). Iterative simulations are carried out until small enough errors are obtained. Ideally, the resulting system does not require any actuation torque to perform the motion. However, to stop the system in the pick-and-place positions, actuators counteracting the springs or mechanical breaks are needed.

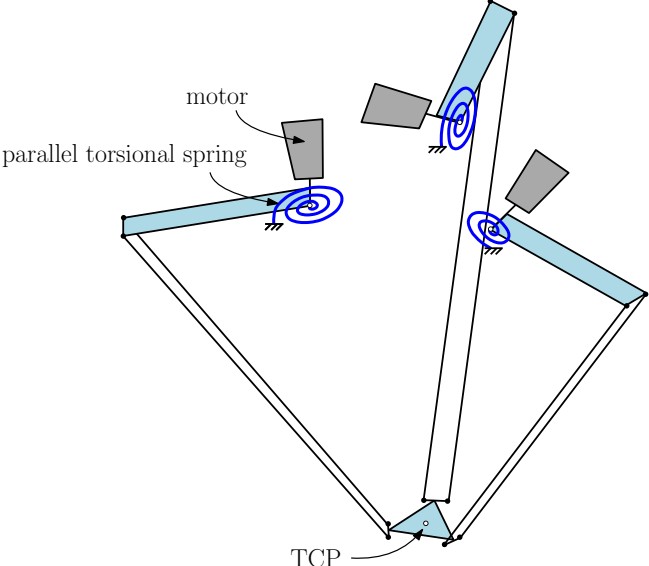

**Figure 7.** Delta robot equipped with parallel torsional springs.

### 3.4. Periodic Trajectory Learning

A mechanical system can move using, as a reference trajectory, the one that it naturally generates. Proper tools, such as adaptive oscillators, can be used to learn the periodic trajectories and adopt them as reference, especially for robot locomotion and dynamic walking [109–111]. In this manner, the reference trajectory is synchronized with the resonant frequency of the system, which results in energy saving. A tool capable of obtaining smooth and lag-free estimates of the frequency and phase of an external quasi-periodic signal is given by the adaptive frequency oscillators (AFO), introduced by L. Righetti et al. [112,113]. AFO are adopted in the literature for the estimation of cyclical movements, especially for rehabilitation purposes and in robots performing quasi-periodic motions. However, the convergence and optimality of these methods are in general not ensured [110,111]. Adaptive oscillators are used in central pattern generator to learn a specific rhythmic pattern, by synchronizing the reference trajectory with the resonant frequency of the robotic system and earning energy efficiency. For example, in Reference [109], Buchli et al. present a 4-DOF spring-mass hopper with a controller based on adaptive frequency Hopf oscillators, which adapts to general, non-harmonic signals. The adaptive oscillator adapts to the properties of the mechanical system, in particular to its resonant frequency.

In order to tune the frequency and shape of the cyclic natural motion for energy efficient, novel oscillators, that is, the adaptive natural oscillators, are introduced. M.R.S. Noorani et al. present in Reference [106] an adaptive frequency non-linear oscillator for energy efficiency that exploits the resonant mode in a leg-like mechanical system called stretchable pendulum. The system is a simple oscillating mass-spring mechanism that interacts with the ground during its oscillation. A Hopf non-linear oscillator is placed in the feedback loop and its frequency tracks the resonance frequency of the mechanical system. The system not only earns energy efficiency but has also the ability to adapt with a changing environment.

M. Khoramshahi et al. and R. Nasiri et al. present in References [83,105] a linear and a non-linear adaptive natural oscillator, ANO and NANO, respectively. These tools are capable of tuning the frequency and the shape of cyclic motions for energy efficiency and ensure optimality and convergence. Moreover, they are built upon the adaptive frequency oscillators but, in contrast to AFO that adapts to the frequency of an external signal, ANO adapts the frequency of reference trajectory to the natural dynamics of the system (Figure 8). In Reference [105], the efficiency of ANO is shown in the simulations of a hopper leg and of a compliant robotic manipulator performing a cyclic task. Furthermore, experimental results of a 1-DOF joint with variable compliance (Figure 9) show the feasibility of the approach, exploiting the natural dynamics and reducing the consumed energy. In Reference [83], the non-linear adaptive natural oscillator (NANO), adopted to optimize the shape of reference trajectories, is presented. The oscillator ensures stability, convergence and optimal solution. Three robotic models are investigated in simulation: the pendulum, the adaptive toy and the hopper-leg, showing the feasibility of the proposed approach to achieve energy efficiency.

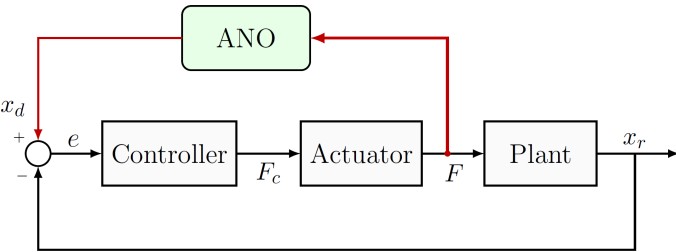

**Figure 8.** The adaptive natural oscillator proposed by M. Khoramshahi et al. [105] to exploit natural dynamics. The oscillator is employed as a pattern generator and the applied force is used as feedback.

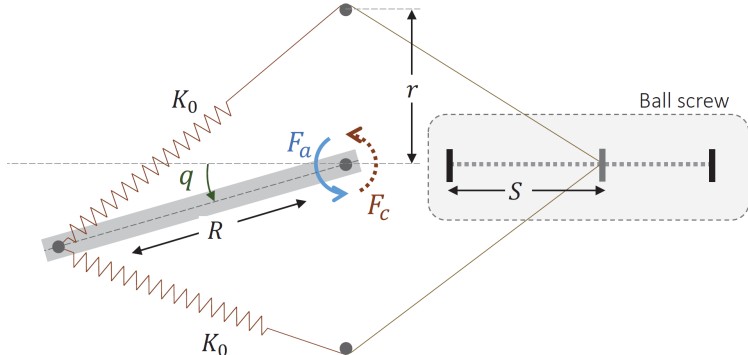

**Figure 9.** The 1-DOF mechanism with variable compliance presented by M. Khoramshahi et al. [105]. The linear spring acts as a parallel rotational spring at the joint and its stiffness can be tuned by controlling the ball screw mechanism.

## 4. Design Optimization with Natural Motion

In the previous sections, we have analyzed the mechanical design of both robotic and mechatronic systems for natural motion and of desired natural dynamics to ensure energy efficiency. It should be noted that, in the majority of the works, the exploitation of natural motion is strictly related to the definition of an optimal design problem. Indeed, spring parameters (with variable or non-variable compliance and linear or non-linear behavior) as well as trajectory parameters are designed using an optimization strategy. In this section, we provide an overview on the optimization problems adopted for the design of trajectory and spring parameters. Table 2 reports a summary on the optimal design problems, showing design variables, objective and constraint functions and algorithms adopted in the reviewed works. Furthermore, in the last column of the table the results obtained in terms of energy (or torque) efficiency are reported.

Before analyzing the optimization problems in detail, it is necessary to make a distinction between works in which the design formulation includes spring parameters, works in which it considers trajectory parameters or both. In problem formulations in which the spring parameters are to be determined, both stiffness and equilibrium position are typically design variables. The second design formulation finds the optimal values for the trajectory and task time, frequency or coefficients of the motion profile are design variables. Another formulation is the concurrent design optimization of spring parameters and trajectory properties. This is the most complex optimization formulation to be implemented, since the properties of elastic elements usually influence the trajectory and *vice versa*. Despite the complexity, this formulation allows for a high degree of design flexibility and, therefore, potential to further reduce energy consumption.

Several methods are used to solve the optimal design problem with natural motion. Due to their efficiency with convex problems, the majority of the works about natural motion utilize gradient-based optimization algorithms. Due to their use of gradients and also use of Hessians, these algorithms are referred to as first and second order algorithms. J.P. Barreto et al. [81] adopt the gradient-based trust-region-dogleg algorithm on an unconstrained problem, whereas H.J. Bidgoly et al. [73], P. Boscariol et al. [89] and R.B. Hill et al. [80] use MATLAB's gradient-based *fmincon* function. R. Fabian et al. [40], as well as M. Khoramshahi et al. [65] and R. Nasiri et al. [83] adopt a gradient descent algorithm. The unconstrained gradient-based *shooting method* was used by R.B. Hill et al. in Reference [91]. N. Schmit et al. [61,102] adopts the sequential quadratic programming (SQP) algorithm. A gradient-free method is implemented in Reference [103], where a downhill simplex algorithm is considered. In some cases, an analytical solution is found to the optimal design problem, avoiding the cost of setting up and solving numerical optimization problems. These papers include References [79,86,96,98,99].

**Table 2.** Optimal design formulation with natural motion.

| Ref. | 1st Author | Year | Design Variables | Objective Function | Optimization Constraints | Algorithm | Results |
|---|---|---|---|---|---|---|---|
| **Spring parameters** | | | | | | | |
| [81] | Barreto | 2018 | stiffness and equilibrium position | position and velocity errors | *bounded unconstrained* | trust-region-dogleg | 100% of torque reduction (numerical) |
| [89] | Boscariol | 2017 | stiffness and location of spring connecting points | torque norm | *bounded unconstrained* | MATLAB *fmincon* | 47% (numerical) and 32% (experimental) of peak torque reduction |
| [90] | Carabin | 2019 | stiffness and equilibrium position | electrical energy | *bounded unconstrained* | second-order NLPQLP | up to 70% of energy reduction (numerical) |
| [91] | Hill | 2019 | coordinates of variable stiffness spring | position and velocity errors | *bounded unconstrained* | *shooting method* | 53% of input torques reduction (numerical) |
| [40] | Fabian | 2018 | configuration of variable stiffness actuator | maximum absolute energy per cycle | *bounded unconstrained* | gradient descent | - |
| [65] | Khoramshahi | 2014 | stiffness and equilibrium position | absolute mechanical work | *unbounded* | analytical solution | 75% of energy reduction (numerical) |
| [98] | Uemura | 2008 | stiffness | torque norm | *unbounded* | control method | up to 55% of torque reduction (numerical) |
| [99] | Uemura | 2009 | stiffness | torque | *unbounded* | control method | 90% of torque reduction (numerical) |
| [86] | Velasco | 2013 | stiffness and equilibrium position | squared-power; squared-torque | task position, velocity and acceleration | analytical solution | up to 90% of cost function reduction (numerical) |
| [79] | Velasco | 2015 | stiffness profile | squared-torque and force needed for stiffness variation | task position, velocity and acceleration | analytical solution | 17% of cost function reduction (experimental) |
| **Trajectory parameters** | | | | | | | |
| [73] | Bidgoly | 2017 | coefficients of torque profile | weighted sum of actuator torque and complexity of torque profiles | end-effector path, physical limitations on joints and motion periodicity | MATLAB interior-point | up to 99.9% of actuator cost reduction (numerical) |
| [82] | Mirz | 2018 | motion profile | energy | *bounded unconstrained* | optimal control theory | up to 84% of energy reduction (numerical) |
| [58] | Nasiri | 2017 | non-linear compliance parameters | actuation force | *bounded unconstrained* | Newton gradient descent | 100% (numerical) and 24% (experimental) of energy reduction |
| **Spring and trajectory parameters** | | | | | | | |
| [103] | Barreto | 2017 | stiffness, equilibrium position and operating time | task time and position errors | *bounded unconstrained* | Nelder-Mead direct search | up to 68% of energy reduction (numerical) |
| [80] | Hill | 2018 | variable stiffness spring and trajectory parameters | power losses of joints and variable stiffness spring motors | *bounded unconstrained* | MATLAB *fmincon* | 48% of energy reduction (numerical) |
| [56] | Iwamura | 2011 | stiffness, equilibrium position and operating time | energy | *bounded unconstrained* | optimal control theory | up to 100% of energy reduction (numerical) |
| [100] | Iwamura | 2016 | stiffness, equilibrium position and operating time | energy | *bounded unconstrained* | optimal control theory | 94% of energy reduction (experimental) |
| [101] | Scheilen | 2009 | stiffness, equilibrium position and operating time | energy | *bounded unconstrained* | optimal control theory | up to 100% of energy reduction (numerical) |
| [102] | Schmit | 2012 | trajectory and torque profiles of non-linear springs | torque | function trajectory parameters at starting and ending time | SQP | up to 99% of actuators torque reduction (numerical) |
| [61] | Schmit | 2013 | trajectory and torque profiles of non-linear springs | torque | stiffness characteristic | SQP | up to 95% of actuators torque reduction (numerical) |
| [96] | Shushtari | 2017 | stiffness and task frequency | total energy | *bounded unconstrained* | analytical solution | - |

An alternative approach is to tackle the problem with an optimal control theory approach. This is used for the optimal design problems by M. Iwamura et al. [56,100], by C. Mirz et al. [82], who along with W. Scheilen et al. [101] implement the Pontryagin's minimum principle (see Section 3.2.1).

Most of the reviewed works adopt objective functions based on actuators torque, on consumed energy or power. A few works use objective functions based on task time and positions [103], position and velocity of the end-effector [81] or position and velocity errors [91]. Results shown in the last column of Table 2 indicate that all the reviewed approaches for the optimal design problems in natural motion consistently reduce energy consumption.

The vast majority of papers utilize gradient-based optimization algorithms. Despite the great efficiency of these algorithms, the validation is typically carried out on simple test cases (as reported in Table 3) with low number of design variables and a convex objective functions. This review did not result in any research using genetic algorithms or evolutionary strategies employed in the field of natural motion. Although, these algorithms are less efficient, they can often handle non-convex problems, discrete design variables and noisy system functions.

**Table 3.** Mechanical systems employed for the numerical and experimental validations, grouped together with respect to their kinematic structure. The check-marks on the second and third columns indicate whether gravity is considered and whether experimental tests are performed, respectively.

| Mechanical System | Gravity | Experimental Tests | Ref. | 1st Author | Year |
|---|---|---|---|---|---|
| 1-DOF revolute joint | | | [7] | Boscariol | 2017 |
| | | ✓ | [40] | Fabian | 2018 |
| | | ✓ | [105] | Khoramshahi | 2017 |
| | ✓ | ✓ | [58] | Nasiri | 2017 |
| | ✓ | | [56] | Iwamura | 2011 |
| | ✓ | | [77] | Uemura | 2007 |
| | ✓ | | [98] | Uemura | 2008 |
| | ✓ | | [86] | Velasco | 2013 |
| | ✓ | ✓ | [79] | Velasco | 2015 |
| 2-DOF planar robot (RR) | ✓ | | [40] | Fabian | 2018 |
| | ✓ | | [56] | Iwamura | 2011 |
| | | ✓ | [100] | Iwamura | 2016 |
| | ✓ | | [104] | Kashiri | 2017 |
| | ✓ | ✓ | [94] | Matsusaka | 2016 |
| | | ✓ | [93] | Matsusaka | 2016 |
| | ✓ | | [58] | Nasiri | 2017 |
| | | | [64] | Schiehlen | 2005 |
| | | | [95] | Schiehlen | 2005 |
| | | | [101] | Schiehlen | 2009 |
| | ✓ | | [96] | Shushtari | 2017 |
| | ✓ | | [77] | Uemura | 2007 |
| | ✓ | | [97] | Uemura | 2008 |
| | ✓ | ✓ | [99] | Uemura | 2009 |
| | | | [78] | Uemura | 2014 |
| | | ✓ | [86] | Velasco | 2013 |
| | | | [79] | Velasco | 2015 |
| 2-DOF planar robot (PP) | ✓ | | [96] | Shushtari | 2017 |
| 3-DOF SCARA | | ✓ | [16] | Goya | 2012 |
| 3-DOF planar robot (RRR) | ✓ | | [73] | Bidgoly | 2017 |
| 3-DOF spatial robot (RRR) | ✓ | | [102] | Schmit | 2012 |
| | ✓ | | [61] | Schmit | 2013 |
| 3-DOF spherical robot (RRP) | ✓ | | [96] | Shushtari | 2017 |
| 4-DOF spatial robot (RRRR) | ✓ | | [73] | Bidgoly | 2017 |
| Slider-crank mechanism | | | [95] | Schielen | 2005 |
| Five-bar linkages | | | [103] | Barretto | 2017 |
| | ✓ | | [80] | Hill | 2018 |
| | ✓ | | [91] | Hill | 2019 |
| | | | [82] | Mirz | 2018 |
| Delta-2 robot | ✓ | | [90] | Carabin | 2019 |
| Delta robot | ✓ | | [81] | Barretto | 2018 |

As it is the goal to consider complex mechanical systems and expanded design problems, the optimization formulation will require a higher number design variables including the parameters

of the motion law. This may lead to non-convex optimization problems for which gradient-based algorithms are not suitable. Thus, the proper optimization formulation, the system equations and corresponding choice of optimization algorithm will play a critical role moving forward with the optimal design with natural motion.

## 5. Discussion

In Section 2, we analyzed several approaches to perform a cyclic task with a robotic system by exploiting its the natural motion, that is, a motion mainly due to the transformation of the potential energy stored by elastic elements into kinetic energy. The application of the natural motion is demonstrated to be beneficial in terms of energy consumption on different mechanical systems (see Table 3) by means of simulation or experiments. From the table, it is evident that most of the methods are validated on very simple test cases, that is, planar systems with one or two revolute joints. In all the cases, systems with rigid-links are considered, so there are no examples of robotic systems with flexible links or fully compliant mechanisms on which the natural motion is applied. Considering the increasing interest in these kind of systems, the effect of link flexibility for the exploitation of the natural motion should be investigated, since link flexibility affects the system motion [114]. This could be taken into account in two ways—controlling the resulting unwanted vibrations or making them part of the natural motion trajectory.

For the practical implementation of the natural motion in industrial application, it is important that the robot can perform multi-point trajectories. Indeed, pick-and-place operations are typically carried out between points belonging to two areas, that is, The pick and the place positions are not always the same but can be any position in these area. The majority of the works discussed in Section 2 neglect such an aspect and only consider point-to-point trajectories. Multi-point trajectories are taken into account in References [16,80,82,91,93,96]. Among these, H. Goya et al. [16] and C. Mirz et al. [82] use spring parameters that are optimized for the center of the interested areas to move the system following a multi-point trajectory. M. Shushtari et al. [96] also consider one constant stiffness, whose value results from an optimization problem that takes into account the whole multi-point trajectory. Conversely, K. Matsusaka et al. [93] and R.B. Hill et al. [80,91] optimize the elastic elements for each segment of the trajectory by changing the stiffness values when the robot is stopped in a pick or in a place position. If small pick-and-place areas are considered, the energetic savings using constant stiffness springs is advantageous with respect to the stiff case and even comparable to the varying stiffness (considering the energy cost to change stiffness) [93], despite being a simpler approach. However, when the points of the multi-point trajectory are far from the averaged values, based on which the spring is optimized, such an approach does not work anymore. It becomes difficult to accurately follow the trajectory (because of the elastic forces to be counteracted) and the energy saving are minimal or even null. Therefore, variable stiffness springs become necessary.

The main goal of the application of the natural motion approach is to save energy. All the reviewed methods allow to improve the energy efficiency of a robotic systems, however, it is almost impossible to assert which one outperforms the others to reach such a goal. A benchmark case, with which to perform a comparative analysis between the different methods, is lacking. The benchmark is meant not only in terms of test case but above all in terms of performance evaluation. All the authors present a percentage of energy savings with respect to the stiff case (see the column of results of Table 2) but someone refers to electrical energy others to mechanical one. Most of the contributions neglect an estimate of the losses or the energy consumed, for example, to change stiffness, to activate the breaks. Additionally, the use of different objective functions (e.g., mechanical torque, mechanical energy, electrical energy) does not allow for a proper comparison, since the use of different objective function can lead to very different results as demonstrated by A. Velasco et al. [86].

Generally speaking, the use of optimized trajectories and variable stiffness springs is the more promising approach, since it leads to a trajectory that better fits the system dynamics and allows for flexibility in task execution (point-to-point or multi-point trajectories, different task frequency). On the

other hand, methods based on fixed trajectories and constant springs are typically easier to set-up and implement.

## 6. Conclusions

In this paper we presented a review, a classification and a discussion of several approaches that adopt the concept of natural motion to enhance the energetic performance in robotic and mechatronic systems. In the first part of the paper, we identified the physical requirements that a system has to fulfill to exploit the natural motion and we discussed the technical possibilities to modify it, if necessary. To this end, the configurations in which compliant elements are installed at the joints of the mechanisms (i.e., serial and parallel, with variable and non-variable stiffness) were introduced and compared. Although a serial arrangement seems to be more convenient in terms of energy savings, most of the works deal with compliance mounted in parallel with the main actuators, because of the ease of installation and control.

In the second part of the paper, we classified the approaches related to natural motion on the basis of the trajectory followed by the system: given trajectory, optimized trajectory, free-vibration response and period trajectory learning. Moving from the first category towards the last, the resulting motion gets closer to the system dynamics. Indeed, in the last case, also known as natural dynamic exploitation, the periodic or cyclic task is designed to match the response naturally generated by a given system. In all the other cases, a modification of the system is typically required (natural dynamic modification) so that its dynamic behavior matches a desired task. Regardless the trajectory followed, the results are that, in general, the added compliance parameters can be optimized offline (in the case of non-variable compliance) or can be tuned online (if the compliance is variable). To conclude, methods based on optimized trajectory and variable stiffness springs are most promising. In fact, they allow to approximate quite well the system dynamics (and hence to reduce energy consumption), while preserving task flexibility.

Furthermore, we presented an analysis of optimal design problems in natural motion, since the implementation of natural motion is often related to the definition of an optimization problem. Numerical and experimental results, expressed as percentage reduction of required energy or torque, show that the adoption of natural motion techniques allows to highly increase energy efficiency.

Future developments in the field of natural motion for energy saving in robotic and mechatronic systems should be toward the creation of a benchmark case to better understand the improvements and the potentialities of the different strategies. Further investigations would be necessary in regards to applicability to more complex systems, for example, spatial mechanisms and robots with different kinds of joints. Furthermore, the concept of natural motion could also be extended to robots and mechanisms with flexible links or fully compliant joint-less systems. Finally, the natural motion approach is expected to be applied in more scenarios in both industrial and academic research, where novel challenges could be addressed to maximize its potentialities.

**Author Contributions:** All authors discussed and commented on the manuscript at all stages. More specifically, L.S. and I.P. collected the related literature, conducted the analysis and completed the draft writing under the supervision of A.G. and R.V.; E.W., A.G. and R.V. contributed to the revision of the paper structure and the presentation style, as well as the proofreading of the paper.

**Acknowledgments:** This work was partially supported by the Free University of Bozen-Bolzano funds within the project TN2803: "Mech4SME3—Mechatronics for Smart Maintenance and Energy Efficiency Enhancement". This work was supported by the Open Access Publishing Fund provided by the Free University of Bozen-Bolzano.

**Conflicts of Interest:** The authors declare no conflict of interest.

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
