# Peer review of "Natural Motion for Energy Saving in Robotic and Mechatronic Systems"

_applsci, doi:10.3390/app9173516_

Round 1

Reviewer 1 Report

The manuscript presents a review of mechatronics systems that exploit the concept of natural motion for energy saving purpose. After an initial discussion of mechanical designs for natural motion, the core of the paper deals with a description of many methods for natural motions proposed in the literature, by categorizing them on the basis of the trajectory they follow. The paper concludes with a discussion on the proposed solutions and still open issues.
My general opinion on the manuscript is good. The authors offer a way to classify numerous references provided in the document, which give a comprehensive view of current solutions for natural motion methods. However, my major remark to the manuscript is the 'flatness' of the presentation. In my view the work could be made more appealing by enriching the text with pictures of most important mechanisms as well as graphical elements (for instance diagrams) explaining basic concepts presented in Section 3.  It is also my opinion that the most interesting part of the paper is the final discussion, where the authors provide a comparison summarizing pros and cons of the proposed solutions. In this context, although authors claim the lack of a common benchmark for comparing energy-saving properties, basing on the authors' expertise, a more quantitative comparison might still be attempted. It could be valuable, especially when considering that it is the main goal of natural motion systems.

Minor remarks:

+ Figure 1: labels a) and b) are missing
+ Line 204: can be referred as; Suggested: can be referred to as
+ Line 225: Such a results; Suggested: Such a result
+ Line 336: A method that allows...; This sentence lets suggests (at least to me) that a method that allows to exploit stiffness variability during a task is a peculiarity of [76]. In practice all methods described above share the same feature
+ Line 596: neglects; Suggested: neglect
+ Line 615: refer to; Suggested: refers to
+ Table 3: The caption should better explain the contents of the table (as it is not explained in the text). What does gravity means? A mechanism featuring gravity-balancing?

Reviewer 2 Report

   In the paper, a comprehensive review is conducted regarding the approaches that adopt the concept of natural motion to enhance the energetic performance in robotic system. The topic is important. The problems are clearly addressed. The classification, explanation, comparison and discussion are reasonable. The literature survey is complete. However, the reviewer would like to give the following suggestions:

Since large amount of examples are applied in the paper, more figures and illustrations are suggested. The number of figures and tables is too less in the paper. It is difficult to image the configuration of the examples given in the paper. Since it is for energy saving, suggesting increasing the introduction on this part. Compared with the rest part, the address on energy saving is less. It is difficult to see how the specific system helps to save the energy.
